# Synergistic Effects of Venetoclax and Daratumumab on Antibody-Dependent Cell-Mediated Natural Killer Cytotoxicity in Multiple Myeloma

**DOI:** 10.3390/ijms221910761

**Published:** 2021-10-05

**Authors:** Ayano Nakamura, Susumu Suzuki, Jo Kanasugi, Masayuki Ejiri, Ichiro Hanamura, Ryuzo Ueda, Masao Seto, Akiyoshi Takami

**Affiliations:** 1Department of Internal Medicine, Division of Hematology, Aichi Medical University School of Medicine, Nagakute 480-1195, Japan; ayano.n@aichi-med-u.ac.jp (A.N.); jo@aichi-med-u.ac.jp (J.K.); hanamura@aichi-med-u.ac.jp (I.H.); seto_masao@aichi-med-u.ac.jp (M.S.); takami-knz@umin.ac.jp (A.T.); 2Research Creation Support Center, Aichi Medical University, Nagakute 480-1195, Japan; 3Department of Tumor Immunology, Aichi Medical University School of Medicine, Nagakute 480-1195, Japan; uedaryu@aichi-med-u.ac.jp; 4Department of Pharmacy, University Hospital, Aichi Medical University, Nagakute 480-1195, Japan; ejiri.masayuki.107@mail.aichi-med-u.ac.jp

**Keywords:** multiple myeloma, daratumumab, venetoclax, BCL-2, ADCC, NK cell

## Abstract

The prognosis of multiple myeloma (MM) has drastically improved owing to the development of new drugs, such as proteasome inhibitors and immunomodulatory drugs. Nevertheless, MM is an extremely challenging disease, and many patients are still refractory to the existing therapies, thus requiring new treatment alternatives. Venetoclax is a selective, orally bioavailable inhibitor of BCL-2 that shows efficacy in MM not only as a single agent but also in combination therapy, especially for MM patients with translocation t(11;14). However, many patients are refractory to this drug. Here, we treated the MM cell lines KMS12PE and KMS27 with a combination treatment of venetoclax targeting BCL-2 and daratumumab targeting CD38 to evaluate the synergistic cytotoxicity of these drugs in vitro. MM cell lines were co-cultured with natural killer (NK) cells at an effector:target ratio of 0.3:1 in the presence of serial concentrations of daratumumab and venetoclax, and the resulting apoptotic MM cells were detected by flow cytometry using annexin V. These results indicated that the antibody-dependent cell-mediated NK cytotoxicity was enhanced in KMS12PE and KMS27 cells harboring t(11;14) with a high BCL-2 expression, suggesting that the combination treatment of venetoclax and daratumumab should be especially effective in patients with these characteristics.

## 1. Introduction

The clinical introduction of proteasome inhibitors (PIs), such as bortezomib, and immunomodulatory drugs (IMiDs), such as thalidomide and lenalidomide, significantly improved the prognosis for multiple myeloma (MM) patients. Treatment with monoclonal antibodies has been shown to be effective for MM patients refractory to PIs and IMiDs, which was demonstrated by daratumumab targeting CD38 and elotuzumab targeting SLAMF7. Although many patients are currently successfully treated and achieve a complete response, many experience relapse in the long term and require additional treatment options [1].

The BCL-2 family of proteins is composed of crucial apoptosis regulators [2]. These include pro-survival and pro-apoptotic proteins, and interactions between these two classes of proteins are crucial in making cell fate decisions [2]. BCL-2 proteins are classified into three major groups according to structure and function: anti-apoptosis proteins include MCL-1, BCL-2, and BCL-XL; multi-domain pro-apoptotic proteins include BAX and BAK1; and BH3-only proteins include BIM, BID, BAD, and NOXA [2].

Anti-apoptosis proteins are important therapeutic targets [2] and are targeted by venetoclax (also known as ABT-199), a selective, orally bioavailable and BH3 mimetic inhibitor with high affinity to BCL-2 but not to BCL-XL or MCL-1 [2]. Venetoclax is regarded as a new option for the treatment of MM, with a mechanism of action different from that of PIs and IMiDs; it is currently being evaluated through clinical trials in patients with MM [2]. Venetoclax induced apoptosis in human MM cell lines and in vitro primary samples collected from patients with MM, especially in those harboring translocation t(11:14) (q13:q32) [3,4], which is present in 15% to 20% of patients with MM [3,4].

In the phase I study of venetoclax monotherapy for relapsed/refractory (R/R) MM, the overall response rate (ORR) was 21%, and 15% of patients achieved a very good response rate (VGPR) or better (NCT01794520) [5]. Within the group of patients carrying t(11;14), the ORR was 40%, and 27% patients achieved a VGPR or better [5]. In this study, patients with high gene expression ratios of *BCL2* to *BCL2L1* and *BCL2* to *MCL1* were more sensitive to venetoclax than patients with low ratios [5].

Venetoclax has shown efficacy for MM not only in monotherapy but also in combination therapy. In a phase Ib study evaluating venetoclax together with bortezomib and dexamethasone in R/R MM, the ORR was 67%, and 42% of patients achieved a VGPR or better (NCT01794507) [6]. In this study, 94% of the patients with high *BCL2* levels achieved ORR, while in those with low *BCL2* levels, the ORR was 59%. Bortezomib inhibits MCL-1 indirectly by stabilizing the MCL-1-neutralizing protein NOXA [7]. In xenograft models resistant to venetoclax that co-express BCL-XL or MCL-1 with BCL-2, this resistance was decreased by bortezomib [8]. Similarly, dexamethasone upregulates the pro-apoptotic BIM and increases its binding to BCL-2, which also results in increased sensitivity to venetoclax [6,9].

Some patients do not respond to this drug, while others show progress after an initial response. The efficacy and safety of a treatment using venetoclax together with daratumumab and dexamethasone (VenDd) is currently being evaluated (NCT03314181) [2]. Bortezomib and dexamethasone show similarities to venetoclax in that they target the BCL-2 family and improve treatment effectivity. The antibody-dependent cell-mediated cytotoxicity (ADCC) apoptotic pathway also activates BID, a BH3-only protein from the BCL-2 family, to induce BAX activation. In an attempt to explore a new approach for the treatment of patients with MM, here we aimed to evaluate, using in vitro techniques, whether the combination treatment of venetoclax and daratumumab increased cytotoxicity. Our results will contribute to future studies on the clinical applications of this type of treatment, including the establishment of the optimal dose for each of these drugs.

## 2. Results

### 2.1. Expression Levels of CD38 in MM Cell Lines

CD38 was highly expressed in KMS12PE and moderately expressed in KMS27 and KM5; in contrast, CD38 expression was low in U266 (Figure 1).

### 2.2. Expression Levels of BCL-2 Family Proteins in MM Cell Lines

BCL-2 family proteins levels varied among the MM cell lines (Figure 2). BCL-2 was more highly expressed in KMS12PE than in KMS27, U266, and KM5. High protein level of BCL-XL was observed in U266; in contrast, BCL-XL protein level was very low in KMS12PE, KMS27, and KM5. As for MCL-1, the protein level was higher in KM5 than in KMS12PE, KMS27, and U266. BCL-2:BCL-XL and BCL-2:MCL-1 ratios were higher in KMS12PE and KMS27 than in U266 and KM5. This observation correspondent to venetoclax sensitivity (Figure 3).

### 2.3. Sensitivity of MM Cell Lines to Venetoclax

The sensitivity of MM cell lines to venetoclax was examined by a WST-1 assay. As shown in Figure 3, the IC50 values of KMS12PE, KMS27, U266, and KM5 for venetoclax were 6.3, 3.5, 10000, and 3100 nM, respectively. KMS12PE and KMS27 were sensitive to venetoclax but KM5 and U266 were resistant.

### 2.4. Synergistic Effects of Venetoclax and Daratumumab on ADCC

In order to compare the cytotoxic activity of venetoclax alone, daratumumab alone, and a combination of both agents, flow cytometry was performed for detecting annexin V. Representative results are shown in Figure 4. In KMS12PE, the annexin V positivity was 19.3% when 2.5 nM venetoclax was used alone, 41.9% when 25 ng/mL daratumumab was used alone, and 64.2% when a combination of 2.5 nM venetoclax and 25 ng/mL daratumumab was used (Figure 4A). On the other hand, in KM5, the annexin V positivity was 15.1% when 2.5 µM venetoclax was used alone, 40.6% when 25 ng/mL daratumumab was used alone, and 31.8% when a combination of 2.5 µM venetoclax and 25 ng/mL daratumumab was used (Figure 4B).

These experiments were repeated three times in each cell line, and the average value with standard deviation (SD) has been shown (Figure 5). A synergistic effect was observed in KMS12PE and KMS27, which are sensitive to venetoclax, but not in U266 and KM5, which are resistant to venetoclax.

The perforin inhibitor concanamycin A (CMA) did not inhibit the cytotoxicity of venetoclax alone but clearly inhibited the synergistic cytotoxicity of daratumumab and venetoclax (Figure 6).

## 3. Discussion

This study showed that venetoclax enhanced the ADCC induced by daratumumab in the MM cell lines that are sensitive to venetoclax. This was especially true for cell lines harboring t(11;14) with high levels of CD38 and BCL-2. KMS12PE [10] and KMS27 [11] are examples of MM cell lines harboring t(11;14) (q13:q32) and a high expression of BCL-2 [10]. The in vitro venetoclax monotherapy was more effective for MM in cell lines with t(11;14) than in those without t(11;14). However, U266 expressed low levels of BCL-2, despite the presence of t(11;14) and resistance to venetoclax. The t(11;14) translocation is a hallmark of mantle cell lymphoma and results in an overexpression of cyclin D1 [12]. The associations between t(11;14) and BCL-2 expression remain poorly understood. A phase I trial for patients with R/R MM revealed the effectiveness of venetoclax in the patients with high *BCL2*:*BCL2L1* or *BCL2*:*MCL1* ratios. In this trial, 38% of the patients with t(11;14) expressed a high *BCL2*:*BCL2L1* gene expression ratio, while only 5 % of the patients without t(11;14) had a high ratio of the gene expression [2,5,9]. Interestingly, KMS12PE and KMS27, which demonstrated high BCL-2:BCL-XL and BCL-2:MCL-1 ratios, were sensitive to venetoclax; in contrast, U266 and KM5, which showed low ratios, were resistant to venetoclax. The results of our in vitro experiments are consistent with the results of this clinical trial.

Our data correlated with those from a previous case report [13] and with the currently reported results of phase I/II trials for patients with MM [14,15]. A case report of a patient with relapsed plasma cell leukemia and t(11;14) treated with a combination therapy of venetoclax, daratumumab, and dexamethasone showed a rapid decrease in tumor burden [13]. Another group reported a phase I/II trial comprising 48 patients with R/R MM that were treated with a combination therapy of venetoclax and daratumumab to evaluate the efficacy and safety of this treatment strategy. Twenty-four MM patients with t(11;14) who were treated with more than one prior line of therapy (including PIs or IMiDs) were administered the combination therapy VenDd. Among these patients, 46% were refractory to PI, 71% were refractory to IMiDs, and 42% were double refractory to PI and IMiDs. The ORR of patients with t(11;14) treated by VenDd therapy was 92%, while that for patients treated with venetoclax (Ven) in combination with daratumumab (D), dexamethasone (d), and bortezomib (V) (VenDVd) therapy was 88%; this demonstrated a significant efficacy of the combination treatment [14,15]. Our data are also useful for establishing the optimal dose for each of the drugs. The effective treatment concentration can be evaluated based on in vitro data using samples obtained from patients.

Our data strongly suggested that natural killer (NK) cells functioned in vitro. Although the efficacy of the combination therapy of venetoclax and daratumumab has already been shown in clinical trials, our data provide significant meaning with the presence of ADCC activity by NK cells in vitro.

The induction of apoptosis by NK cells is one of the mechanisms of action of daratumumab [16]. In vitro data demonstrated that the level of CD38 expression in MM cells is correlated with the ADCC induction by daratumumab [17], and CD38 is produced in both NK and MM cells [18]. In light of these results, we need to further evaluate whether the level of CD38 expression in MM cells affects treatment response after the combination therapy of venetoclax and daratumumab. This should be performed using in vitro assays with samples taken from patients. The number of NK cells rapidly decreased in both the bone marrow and peripheral blood after daratumumab monotherapy [18]. This reduction in NK cells results in cytotoxicity among neighboring NK cells (NK cell fratricide) [19]. NK cells with low or no CD38 expression were resistant to their own cytotoxicity induced by daratumumab in an experiment using the peripheral blood and bone marrow of MM patients or healthy donors [19]. Indeed, these cells demonstrated superior ADCC activity induced by daratumumab than NK cells presenting higher CD38 expression levels [19,20]. Whether CD38 expression is correlated with the therapeutic response to a combination therapy of venetoclax and daratumumab remains to be evaluated.

This study has some limitations. It has been reported that the antimyeloma effect of daratumumab is induced by complement-dependent cytotoxicity (CDC) and antibody-dependent cellular phagocytosis (ADCP), in addition to ADCC as shown in this study [21,22,23]. Therefore, we attempted additional experiments to confirm whether venetoclax could enhance CDC by daratumumab using two CD38-positive myeloma cell lines (KMS12PE [10] and KMS27 [11]), which were used in this study and had revealed daratumumab mediated ADCC. However, daratumumab did not cause CDC in either of the two myeloma cell lines before examining the CDC-enhancing effect of venetoclax, as shown in Appendix A. These results may be supported by a previous report [16] that analysis of tumor cells in patients with myeloma found only half of the patients had daratumumab-induced CDC. Therefore, it remains unclear whether CDC and ADCP could be involved in the synergistic effect of daratumumab and venetoclax on myeloma cells. However, venetoclax specifically inhibits BCL-2 in mitochondrial, causing caspase activation via the mitochondrial apoptotic pathway and exerting cytotoxic activity [24,25,26]. In ADCC by NK cells, the intracellular entry of granzyme through perforin pore formation induces caspase activation via the mitochondrial apoptotic pathway [27]. These suggest that the synergistic effect of ADCC and venetoclax by NK cells via daratumumab on MM cells may have been induced by activation of the mitochondrial apoptotic pathway. In contrast, the antitumor effects of CDC and ADCP are caused by the direct cell-killing effect by lysis through complement pore formation and phagocytosis by macrophages, respectively, and it is considered that the mitochondrial apoptosis pathway is not directly involved [28,29,30,31]. These suggest that venetoclax is unlikely to enhance CDC or ADCP against MM cells via daratumumab. However, these are highly speculative and require further investigation in the future.

In the future, both the characterization of MM samples and the measurement of NK cell activity should be essential aspects for determining a treatment strategy for MM. Analyzing the expression of proteins from the BCL-2 family in MM cells of the patient and predicting the treatment response may be useful for therapeutic decision-making. BH3 profiling is an example of a functional assay that investigates the interactions of BCL-2 family members [32]. These genetic and/or functional assays are useful tools for detecting the initial therapeutic target for the newly diagnosed patient by identifying the pro-survival proteins the tumor cells depend on [4]. Enhancing the therapeutic response by measuring and increasing NK cell activity is a promising treatment strategy, and several protocols for the expansion of NK cell activity are currently being investigated [33].

## 4. Materials and Methods

### 4.1. Drugs and Cell Lines

Venetoclax was purchased from Selleck (Houston, TX, USA). Daratumumab was purchased from Janssen (South Raritan, NJ, USA). The MM cell lines KMS12PE [10], KMS27 [11], U266 [34] and KM5 [35] were preserved at the laboratory of the Division of Hematology, Department of Internal Medicine of Aichi Medical University, in Nagakute, Japan. Cell lines were cultured in Roswell Park Memorial Institute (RPMI)-1640 medium with L-glutamine and sodium bicarbonate (R8758-500ML, Sigma-Aldrich, St. Louis, MO, USA), 10% fetal bovine serum (Thermo Fisher Scientific, Waltham, MA, USA), and 1% penicillin-streptomycin (Thermo Fisher Scientific, Waltham, MA, USA).

### 4.2. Flow Cytometry

MM cell lines were stained using an Alexa Fluor 647 Mouse Anti-human CD38 antibody (BioLegend, San Diego, CA, USA) for CD38 detection and an Alexa Fluor 647 Mouse IgG2a, κ Isotype Ctrl antibody (BioLegend, San Diego, CA, USA) as a control. The stained cells were analyzed using a BD Fortessa cytometer (BD Biosciences, San Jose, CA, USA). Data were analyzed using the FlowJo software package (v. 10; Tree Star, Ashland, OR, USA).

### 4.3. Western Blotting

Aliquots of 1 × 10^6^ cells were lysed in 20 µL of ice-cold protein extraction buffer (10 mM Tris-HCl, 150 mM NaCl, 2 mM ethylenediaminetetraacetic acid [EDTA], and 2 mM 2-mercaptoethanol [pH 7.4]) in 1.5-mL microtubes for 30 min. The soluble fraction was separated by centrifugation at 10,000× *g*; 15 µL of the supernatant was mixed with 5 µL of 4× sample buffer. The mixture was boiled for 3 min at 95 °C, and finally 5 µL was loaded on a 4-12% sodium dodecyl sulfate (SDS)-polyacrylamide gradient gel (Novex NuPAGE SDS-PAGE, Invitrogen, Carlsbad, CA, USA). Proteins were fractionated by electrophoresis at 40 mA for 1 h. The separated proteins were transferred to polyvinylidene fluoride (PVDF) membranes using an iBlot 2 Dry Blotting System (Invitrogen, Carlsbad, CA, USA). The membrane was incubated with 5% skim milk in phosphate-buffered saline (PBS) for 30 min at room temperature and then with 1 µg/mL of anti-BCL-2 antibody (MBL, Nagoya, Japan), 1 µg/mL of anti-BCL-XL antibody (MBL, Nagoya, Japan), 1 µg/mL of anti-MCL-1 antibody (CST, Danvers, MA, USA), or 0.1 µg/mL of anti-β-actin antibody (MBL, Nagoya, Japan) with 1% skim milk in PBS for 1 h at room temperature. After washing six times with PBS for 5 min, the membrane was incubated with horseradish peroxidase-conjugated anti-mouse IgG (ImmPRESS HRP REAGENT PEROXIDASE Anti-Mouse IgG, Vector Laboratories, Burlingame, CA, USA) for 30 min at room temperature and washed again. The membrane was developed using a chemical luminescence system. Specific signals were detected with an Image Analyzer (Amersham, GE Healthcare, Chicago, IL, USA).

### 4.4. WST-1 Assay

MM cell lines were resuspended with culture medium at a concentration of 5 × 10^5^ cells/mL. Cell suspensions (100 µL) were seeded in a 96-well flat-bottom plate and cultured at 37 °C for 48 h in the presence of serial concentrations of venetoclax. In each well, 10 µL of the Cell Counting Kit reagent (CytoSelect WST-1 Cell Proliferation Assay Reagent [Colorimetric], CBL, San Diego, USA) was added and the color reaction was performed at 37 °C for 2 h. Absorbance values were determined using a microplate reader (BIO-RAD, Hercules, CA, USA) at 450 and 620 nm. We calculated cell viability using the formula as follows:$$\mathrm{Cell}\;\mathrm{viability}\;\left(\%\right)=\frac{\mathrm{absorbance}\;\mathrm{of}\;\mathrm{the}\;\mathrm{sample}-\mathrm{blank}}{\mathrm{absorbance}\;\mathrm{of}\;\mathrm{the}\;\mathrm{negative}\;\mathrm{control}-\mathrm{blank}}\times 100$$

The x-axis (on a base 10 logarithmic scale) represents venetoclax concentration and the y-axis shows the percentage of cell viability. The sample wells included cell lines and Cell Counting Kit reagent with venetoclax at the indicated concentrations, while the negative control included the cell lines and Cell Counting Kit reagent without venetoclax. The blank included the medium and Cell Counting Kit reagent without cells.

### 4.5. NK Cell Preparation

Peripheral blood mononuclear cells (PBMCs) obtained from a healthy donor were separated by a density gradient at 1500 rpm for 30 min using Ficoll-Paque PLUS (GE Healthcare, Chicago, IL, USA). NK cells were separated using anti-CD56 antibody-coated magnetic micro beads in an autoMACS Pro Separator (Miltenyi Biotec, Bergisch Gladbach, Germany) and expanded using an NK Cell Activation/Expansion Kit (Miltenyi Biotec, Bergisch Gladbach, Germany) according to the manufacturer’s protocol.

### 4.6. Annexin V Assay

The MM cell lines KMS12PE, KMS27, U266, and KM5 were used as target cells for NK cell-mediated ADCC activity assessment using an annexin V assay. NK cells were co-incubated with KMS12PE or KM5 at a 0.3:1 effector: target ratio (E:T) for 18 h in the presence of serial concentrations of daratumumab and venetoclax and were analyzed by flow cytometry. These cells were stained with anti-CD56-PE (BioLegend, San Diego, CA, USA), a specific cell surface marker for NK cells and annexin V (BioLegend, San Diego, CA, USA), which is specific for apoptosis. The stained cells were analyzed with a BD Fortessa cytometer (BD Biosciences, San Jose, CA, USA) and data were analyzed using the FlowJo software package (v. 10; Tree Star, Ashland, OR, USA). Cytotoxicity was calculated as follows: cytotoxicity (%) = [(the measured value at each daratumumab concentration with or without venetoclax)—(the measured value when no daratumumab or venetoclax was used)]/[100—(the measured value when no daratumumab or venetoclax was used) × 100]. The x-axis represents the concentrations of daratumumab and venetoclax whereas the y-axis shows the cytotoxicity percentage.

### 4.7. Perforin Inhibition by CMA

CMA was used as a perforin inhibitor to investigate the downstream effector mechanisms of the combination treatment of daratumumab and venetoclax. KMS12PE cells were cocultured with NK cells, daratumumab, and venetoclax in the presence of 10 nM CMA for 18 h, and cytotoxicity was analyzed by flow cytometry with annexin V following the method described in 4.6.

### 4.8. Statistical Analysis

Differences between the two groups were examined via the Student’s t-test.

## Figures and Tables

**Figure 1 ijms-22-10761-f001:**
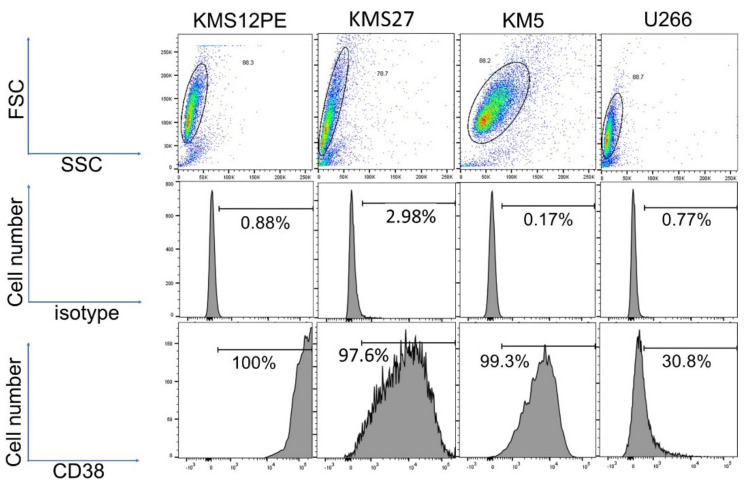
Flow cytometry for CD38 in MM cell lines. The upper panels show cytograms; the x-axis represents SSC and the y-axis represents FSC. The circled areas represent viable cells. The middle and lower panels show flow cytometry histograms. The x-axis shows the positivity for the antibodies and the y-axis represents the cell numbers. The middle panels indicate results with isotype control. The lower panels indicate results with Alexa 647-labeled CD38 antibody.

**Figure 2 ijms-22-10761-f002:**
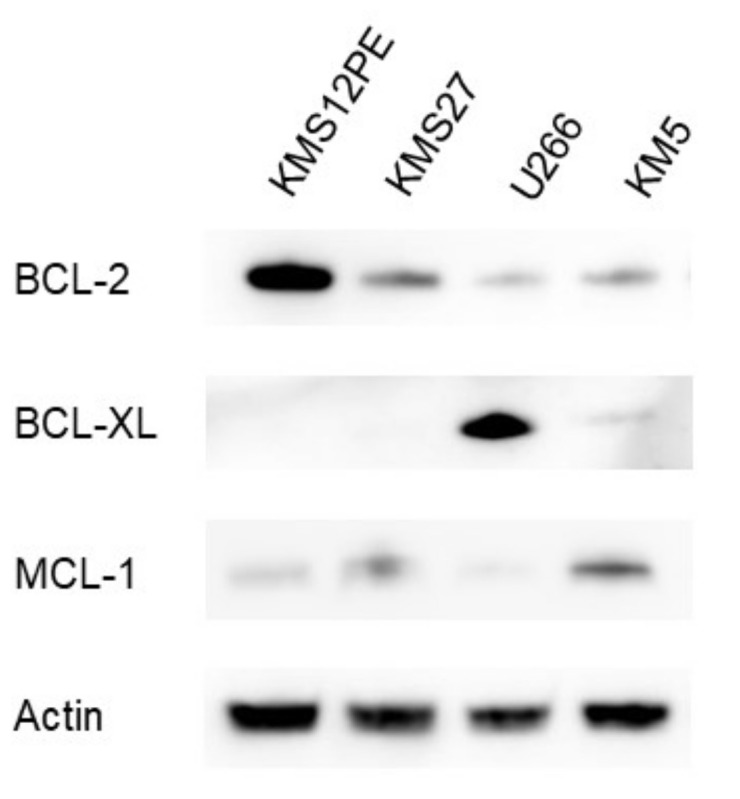
Western blot analyses for BCL-2 family proteins in MM cell lines. Actin was used as a loading control.

**Figure 3 ijms-22-10761-f003:**
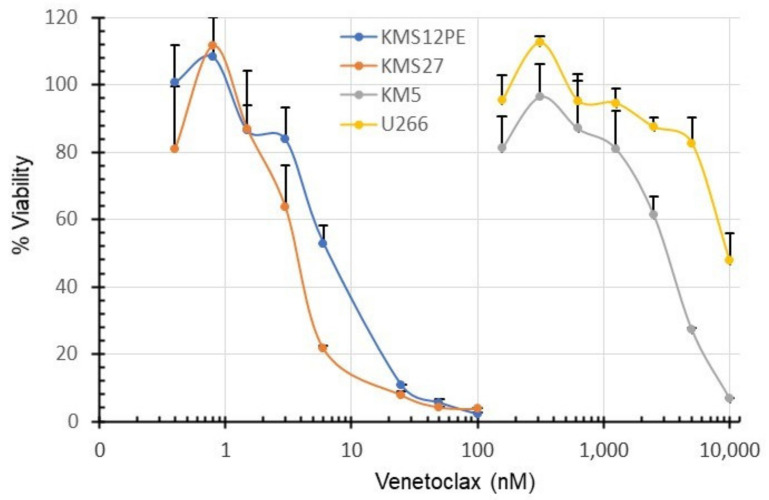
Cytotoxicity of venetoclax to MM cell lines. MM cell lines were treated with venetoclax at the indicated concentrations and cultured for 48 h. The x-axis represents the concentration of venetoclax. The x-axis was formatted to have a base 10 logarithmic scale. The y-axis shows % viability. Viability values on each point is shown as 100% at 0 nM.

**Figure 4 ijms-22-10761-f004:**
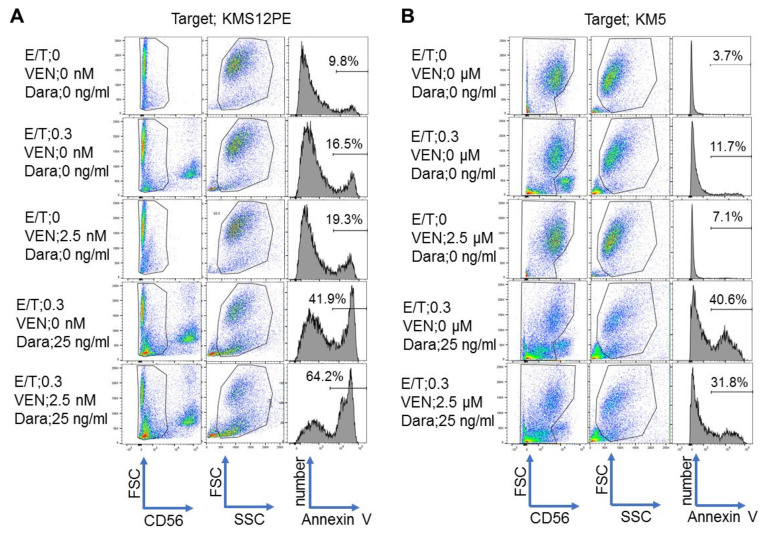
Representative flow cytometric analyses of cytotoxicity by NK cell- mediated ADCC with daratumumab and venetoclax to in MM cell lines using annexin V. The cytotoxicity of combining NK cell mediated ADCC with daratumumab and VEN to in KMS12PE (**A**) and KM5 (**B**) cells was investigated by flow cytometric analyses with annexin V. Cells were gated on forward scatter (FSC) and CD56 to remove NK cells, and the gated cells were separated by forward scatter (FSC) and side scatter (SSC). Annexin V+ cells in gated cells separated by FSC and SSC were detected. The percentage of Annexin V+ cells is indicated in each histogram. Cell treatment conditions are indicated on the left side of each panel. E/T: effector/target, VEN: venetoclax, Dara: daratumumab.

**Figure 5 ijms-22-10761-f005:**
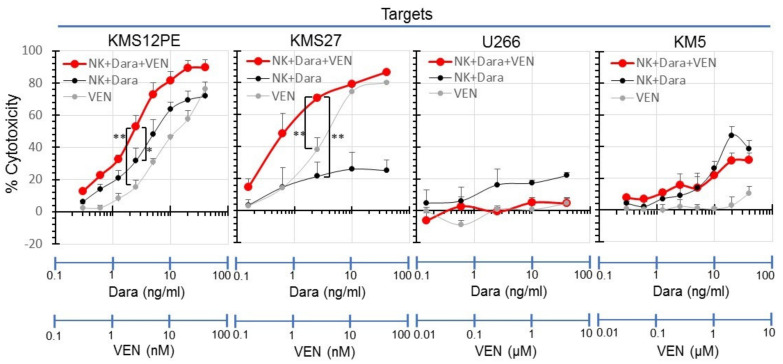
Synergistic cytotoxicity of NK cell mediated ADCC with daratumumab and venetoclax to MM cell lines. Cytotoxicity of NK cell mediated ADCC with daratumumab and venetoclax to MM cell lines, KMS12PE, KMS27, U266 and KM5 was measured by flow cytometry with annexin V. MM cell lines were cocultured with NK cells at effector/target (E/T) ratios: 0.3 with serial concentrations of daratumumab and venetoclax. Cytotoxicity is shown as the mean of three different assays. Red, black, and gray lines indicate the cytotoxicity of NK cells with daratumumab and venetoclax (NK + Dara + VEN), NK cells with daratumumab (NK + Dara), and venetoclax (VEN), respectively. *—differences between groups 0.01< *p* < 0.05, the Student’s t-test. **—differences between groups *p* < 0.01, the Student’s t-test.

**Figure 6 ijms-22-10761-f006:**
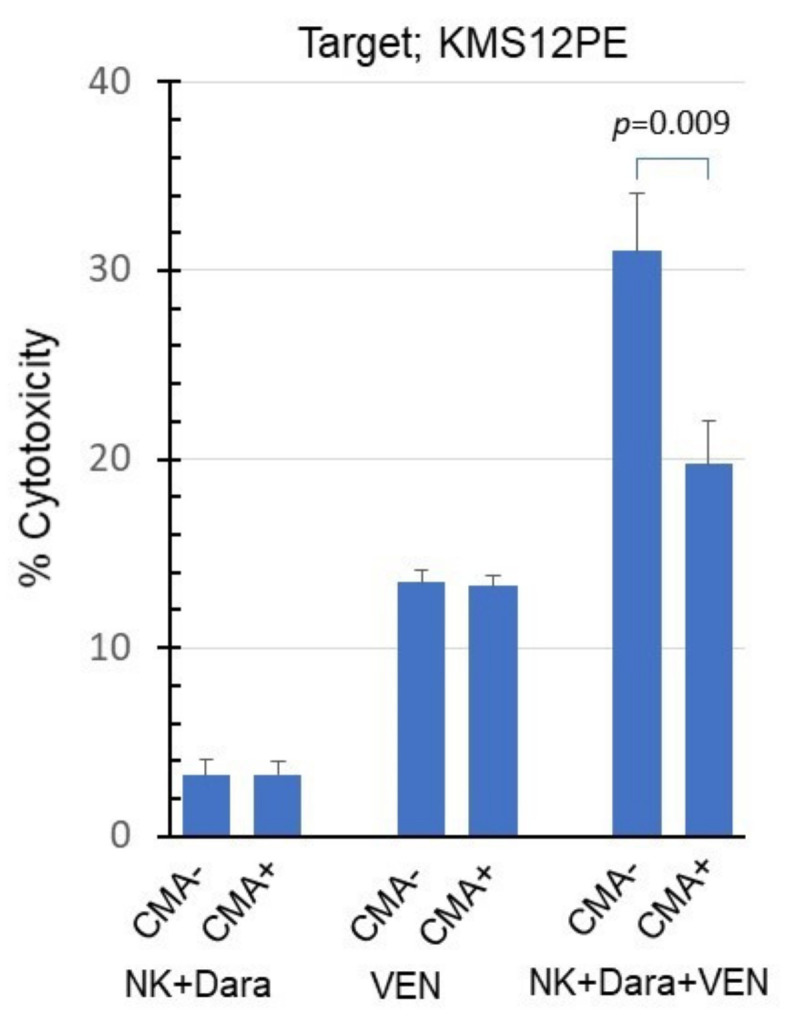
Inhibition of the synergistic cytotoxicity of daratumumab and venetoclax by CMA. KMS12PE was co-cultured with NK cells (E/T = 0.3), daratumumab (2.5 ng/mL), and venetoclax (2.5 nM) in the presence of CMA (10 nM) for 18 h.

## Data Availability

Not applicable.

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
