# Peer review of "Synergistic Effects of Venetoclax and Daratumumab on Antibody-Dependent Cell-Mediated Natural Killer Cytotoxicity in Multiple Myeloma"

_ijms, 2021, doi:10.3390/ijms221910761_

Round 1

Reviewer 1 Report

In this manuscript entitled “Synergistic effect of venetoclax for antibody dependent cell-mediated cytotoxicity by daratumumab in multiple myeloma”, Nakamura et al. assessed the combination efficacy of venetoclax (a Bcl-1 inhibitor) and daratumumab (anti-CD38 mAb) in vitro.

This manuscript has addressed an important clinical aspect; however the following point will need to be improved.

# 1. Perhaps, the quality of the current manuscript can be improved by adding one of the following in vivo experiments.

e.g.  1) xenograft in nude mice treated with/without Dara + Venetoclax, in the absence or presence of anti-asialoGM1 (NK-depleting antibodies).

e.g. 2) xenograft in NRG mice treated with/without Dara + Venetoclax, in the absence or presence of adoptive transfer of NK cells.

If the animal experiments are technically difficult, I would suggest to perform the following 3 in vitro experiments to improve quality and novelty.

Exp 1.  The downstream effector mechanisms of Dara-ADCC + Venetoclax. 

Authors should address whether neutralization of IFNg or perforin inhibitor attenuate the combination efficacy of Dara-ADCC + Venetoclax.    

Exp 2. The impact of Venetoclax on Dara-induced complement-dependent cellular cytotoxicity (CDC) in two MM cell lines. 

Exp 3. The impact of Venetoclax on macrophage antibody-dependent cellular phagocytosis in two MM cell lines.  (e.g. using PMA-differentiated THP1 cell line, clearance of antibody-coated MM cells should be tested in the presence or absence of venetoclax).

# 2.  Results section should be reo-organized.  The current style has multiple sections such as “2.1. Flo cytometry”, “2.2 Western blot”, “2.3. WST1 assay”…. This style is acceptable for methods papers, but not for standard research manuscripts.  Here, authors should specifically describe key results. E.g. “Expression levels of CD38 in myeloma cell lines” ”Expression levels of Bcl-2 protein in myeloma cell lysates” etc…

# 3. Statistical analysis is missing (Figure 5)  

# 4. English style should be re-checked. 

Author Response

Response to Reviewer 1

In this manuscript entitled “Synergistic effect of venetoclax for antibody dependent cell-mediated cytotoxicity by daratumumab in multiple myeloma”, Nakamura et al. assessed the combination efficacy of venetoclax (a Bcl-1 inhibitor) and daratumumab (anti-CD38 mAb) in vitro.

This manuscript has addressed an important clinical aspect; however the following point will need to be improved.

We appreciate your comments, which have provided valuable suggestions that have improved our paper’s quality. Please refer to our point-by-point responses below. We believe that the additional data provided in this resubmission would help enhance understanding of our conclusions. Thank you again for your help.

# 1. Perhaps, the quality of the current manuscript can be improved by adding one of the following in vivo experiments.

e.g.  1) xenograft in nude mice treated with/without Dara + Venetoclax, in the absence or presence of anti-asialoGM1 (NK-depleting antibodies).

e.g. 2) xenograft in NRG mice treated with/without Dara + Venetoclax, in the absence or presence of adoptive transfer of NK cells.

Unfortunately, we could not perform in vivo experiments using a mouse model owing to time constraints. However, your suggestions are very valuable. We would like to plan in vivo experiments based on your suggestions in the near future.

If the animal experiments are technically difficult, I would suggest to perform the following 3 in vitro experiments to improve quality and novelty.

Exp 1.  The downstream effector mechanisms of Dara-ADCC + Venetoclax.

Authors should address whether neutralization of IFNg or perforin inhibitor attenuate the combination efficacy of Dara-ADCC + Venetoclax.   

Thank you for this important suggestion. We investigated the effects of a perforin inhibitor on the combination efficacy of Dara-ADCC + Venetoclax. KMS12PE was co-cultured with natural killer (NK) cells (E/T=0.3), in the presence of daratumumab (2.5 ng/mL), venetoclax (2.5 nM) and a perforin inhibitor, concanamycin A (CMA), for 18 hours. CMA did not inhibit the cytotoxicity of venetoclax in a single dose but reduced the combination efficacy of Dara-ADCC + Venetoclax. Please refer to Figure 6 and the new sentence added on page 5, lines 147-149 in the Results section of the revised manuscript. The procedure was added to page 8, lines 296-300 in the Materials and Methods section of the revised manuscript.

Exp 2. The impact of Venetoclax on Dara-induced complement-dependent cellular cytotoxicity (CDC) in two MM cell lines.

Exp 3. The impact of Venetoclax on macrophage antibody-dependent cellular phagocytosis in two MM cell lines.  (e.g. using PMA-differentiated THP1 cell line, clearance of antibody-coated MM cells should be tested in the presence or absence of venetoclax).

We appreciate your suggestions and completely agree with them because CDC and ADPC are the other important mechanisms of cytotoxicity elicited by daratumumab. Although we attempted to investigate these aspects, we could not complete the experiments due to shortage of time. ADCC and venetoclax elicit cytotoxicity via a mitochondria-dependent apoptosis pathway, whereas CDC or ADPC cytotoxicity is mitochondria independent. It is considered that mechanisms of combination efficacy of ADCC and venetoclax are different from those of CDC and venetoclax or ADPC and venetoclax. Therefore, we focused on enhancement of ADCC by venetoclax in this study. We would like to report the impact of venetoclax on Dara-induced CDC and ADPC in another paper.

# 2.  Results section should be reo-organized.  The current style has multiple sections such as “2.1. Flo cytometry”, “2.2 Western blot”, “2.3. WST1 assay”…. This style is acceptable for methods papers, but not for standard research manuscripts.  Here, authors should specifically describe key results. E.g. “Expression levels of CD38 in myeloma cell lines” ”Expression levels of Bcl-2 protein in myeloma cell lysates” etc…

Thank you for this comment. Following your suggestions, we corrected the style as below.

2.1. Flow cytometry > 2.1. Expression levels of CD38 in MM cell lines

2.2. Western blotting > 2.2. Expression levels of Bcl-2 family proteins in MM cell lines

2.3. WST-1 assay > 2.3. Sensitivity of MM cell lines to venetoclax

2.4. Annexin V assay > 2.4. Synergistic effects of venetoclax and daratumumab on  ADCC

# 3. Statistical analysis is missing (Figure 5) 

Thank you for this comment. T-test results were inserted in revised Figure 5. Please refer to the revised Figure 5 and the new sentences added to Materials and Methods (page 8, lines 302-303).

# 4. English style should be re-checked.

The revised manuscript was edited for language by a language editing company, Editage (www.editage.com). We feel that the revised manuscript is significantly improved over the initial submission.  

Reviewer 2 Report

It is interesting manuscript showing the synergistic effect of daratumumab and venetoclax in MM in vitro. It requires English editing. Moreover, these points should be improved or answered:

Introduction

  • Line 31-32: It is a bit illogic to state that the ‘development’ of proteasome inhibitors or IMIds improved MM patients’ prognosis. Please, change to ‘clinical introduction’ or synonymous…
  • Line 34: abbreviation of proteasome inhibitors (PIs) appears here, but the first mention of them is in line 31 and there the abbreviation should be already mentioned.

Results

  • Line 93, Figure 1 legend, please omit the sentence: Both cell lines are strong positive with CD38. This is interpreted in the results section and does not have to be in the figure legend
  • Line 102, Figure 2 legend please omit the sentence: KMS12PE and KM5. BCL-2 is strongly detected in KMS12PE but weakly detected in KM5, as it repeats the written results. Please state in the figure legend that Actin was used as a loading control.
  • WST-1 assay: please here calculate the respective IC50 values for venetoclax and include in the results section.
  • Figure 3: The dose-response curves for the two cell lines are presented in two graphs, but both graphs have not equal x and y axes scaling (e.g. 100% vs 120% viability and venetoclax concentration is in a range from 0.1-100 uM for KMS12PE and 0.01-10 for KM5. This is all unclear, please represent the dose-response curve in one graph for both cell lines with equal x and y axes scaling and calculate the IC50 values for both cell lines…
  • Lines 117-119, please indicate in which cell lines are these results observed.
  • Figure 4 A and B, please indicate above the representative flow cytometric analyses which cell line is presented in A and which in B.
  • Figure 4: Representative flow cytometric analyses of cytotoxicity by NK cell mediated ADCC with daratumumab and venetoclax to MM cell lines using Annexin V. The authors present here several designs of experiments, which nicely show in KMS12PE (with t(11;14) and high BCL2) synergy of venetoclax with daratumumamb in a presence of NK cells. The NK cells alone, without the presence of treatment, show already a cytotoxic effect on MM cell lines. I wonder, what would be the cytotoxic effect of Dara alone and combined with venetoclax, without the presence of NK cells. This control experiment I miss in this design.

Discussion

  • Please improve the language level

Author Response

Response to Reviewer 2

It is interesting manuscript showing the synergistic effect of daratumumab and venetoclax in MM in vitro. It requires English editing. Moreover, these points should be improved or answered:

We appreciate your suggestions. According to your suggestions, we have revised the language in the text. Please refer to our point-by-point responses below.

Introduction

    Line 31-32: It is a bit illogic to state that the ‘development’ of proteasome inhibitors or IMIds improved MM patients’ prognosis. Please, change to ‘clinical introduction’ or synonymous…

Thank you for your comment. We have changed the term to ‘clinical introduction’.

    Line 34: abbreviation of proteasome inhibitors (PIs) appears here, but the first mention of them is in line 31 and there the abbreviation should be already mentioned.

.

Thank you for pointing this out. We have introduced the abbreviation at the first instance of its mention on line 31. We have deleted ‘proteasome inhibitors’ in line 34.

Results

    Line 93, Figure 1 legend, please omit the sentence: Both cell lines are strong positive with CD38. This is interpreted in the results section and does not have to be in the figure legend

Thank you for your comment. We have deleted this sentence from revised Figure 1. CD38 expression in myeloma cell lines U266 and KM5 was added to the revised Figure 1 in response to the comment from reviewer 3. Along with this, a new sentence and revised Figure 1 legend were added to the Results section. Please refer page 2, line 84 - page 3, line 86 and page 3, lines 89-93.

    Line 102, Figure 2 legend please omit the sentence: KMS12PE and KM5. BCL-2 is strongly detected in KMS12PE but weakly detected in KM5, as it repeats the written results. Please state in the figure legend that Actin was used as a loading control.

Thank you for your comment. We have deleted this sentence and inserted the sentence “Actin was used as a loading control.” in revised Figure 2. Bcl-2 expression in myeloma cell lines, U266 and KM5, has been included in revised Figure 2, in response to the comment from reviewer 3. Expression levels of BCL-XL and MCL-1 has also been included in revised Figure 2. Along with this, new sentences and revised Figure 2 legend were added to the Results section. Please refer page 3, lines 94-101 and 104-105.

    WST-1 assay: please here calculate the respective IC50 values for venetoclax and include in the results section.

    Figure 3: The dose-response curves for the two cell lines are presented in two graphs, but both graphs have not equal x and y axes scaling (e.g. 100% vs 120% viability and venetoclax concentration is in a range from 0.1-100 uM for KMS12PE and 0.01-10 for KM5. This is all unclear, please represent the dose-response curve in one graph for both cell lines with equal x and y axes scaling and calculate the IC50 values for both cell lines…

    Lines 117-119, please indicate in which cell lines are these results observed.

    Figure 4 A and B, please indicate above the representative flow cytometric analyses which cell line is presented in A and which in B.

    Figure 4: Representative flow cytometric analyses of cytotoxicity by NK cell mediated ADCC with daratumumab and venetoclax to MM cell lines using Annexin V. The authors present here several designs of experiments, which nicely show in KMS12PE (with t(11;14) and high BCL2) synergy of venetoclax with daratumumamb in a presence of NK cells. The NK cells alone, without the presence of treatment, show already a cytotoxic effect on MM cell lines. I wonder, what would be the cytotoxic effect of Dara alone and combined with venetoclax, without the presence of NK cells. This control experiment I miss in this design.

We appreciate your comments on WST1 assay and Annexin V assay.

In the WST-1 assay, IC50 values of venetoclax for the cell lines, KMS12PE, KMS27, U266, and KM5 were 6.3, 3.5, 10000, and 3100 nM, respectively. These results have been described in the Results section of the revised text. Please refer page 3, lines 108-110. The dose response curves for each cell line have been presented in one graph. Please refer revised Figure 3 and its legend (page 4, lines 112-114).

For the Annexin V assay, we apologize for the lack of clarity.

1) Lines 117-119 (118-121 of the revised manuscript) describe the results in KMS12PE cells. ‘In KMS12PE,’ was added ahead of this sentence for enhancing clarity. Please refer page 4, line 118 of the revised manuscript.

2) ‘Target; KMS12PE’ and ‘Target; KM5’ were added above Figure 4A and 4B, respectively.

3) Thank you for the important comment on experiment design for the results shown in Figure 4. We checked the cytotoxicity of daratumumab on MM cell lines in the absence of NK cells. However, no cytotoxicity was observed at least in the concentration range of 0.4-40 ng/ml (data not shown). Therefore, we did not investigate the combination efficacy of daratumumab and venetoclax in the absence of NK cells.

Discussion

Please improve the language level

The revised manuscript was edited for language by a language editing company, Editage (www.editage.com). We feel that the revised manuscript is significantly improved over the initial submission. 

Reviewer 3 Report

This is an interesting manuscript that ivestigates the effect of venatoclax in the presence or absence of the anti-CD38 antibody daratumumab on two myeloma cell lines: high and low BCL-2 expressors. State of the art techniques were used.

As expected, the high BCL-2 expressor cell line has proven to be sensitive to the selective BCL-2 inhibitor, while there was  much less sensitivity in the low expressor. Moreover, daratumumab was much more effective in inducing cell death in the venetoclax sensitive cell line and a synergistic effect is proven. No synergism exists in the less BCL-2 sensitive cell line.

The results are clearly presented and the observation seems valid, though generalization would need substantiation of the observed effect on either a couple of more cell lines or rather with freshly isolated surving myeloma cell cultures.

Author Response

Response to Reviewer 3

This is an interesting manuscript that ivestigates the effect of venatoclax in the presence or absence of the anti-CD38 antibody daratumumab on two myeloma cell lines: high and low BCL-2 expressors. State of the art techniques were used.

As expected, the high BCL-2 expressor cell line has proven to be sensitive to the selective BCL-2 inhibitor, while there was much less sensitivity in the low expressor. Moreover, daratumumab was much more effective in inducing cell death in the venetoclax sensitive cell line and a synergistic effect is proven. No synergism exists in the less BCL-2 sensitive cell line.

The results are clearly presented and the observation seems valid, though generalization would need substantiation of the observed effect on either a couple of more cell lines or rather with freshly isolated surving myeloma cell cultures.

We appreciate your thoughtful comments. We have further investigated the synergistic effect using additional cell lines, KMS27 and U266. CD38 expression level was moderate in KMS27 and weak in U266 cells (Please refer revised Figure 1). BCL-2 expression level was weak in both KMS27 and U266 cells (Please refer revised Figure 2). Expression of BCL-XL and MCL-1 has been shown in revised Figure 2. The venetoclax dose response curves obtained with the WST-1 assay for both cell lines have been included in revised Figure 3. KMS27 was sensitive to venetoclax, but U266 was resistant. The synergistic effect of the combination of daratumumab and venetoclax was observed for another venetoclax sensitive MM cell line, KMS27, but not for U266, which is resistant (revised Figure 5). New sentences have been added in the Results and Discussion sections of the revised manuscript. Please refer page 2, line 84-page3, line 86; page 3, lines 94-101; page 5, lines 135-138; and page 6, lines 168-171. In addition, new Figure 6 and its legend were included on page 5, lines 152-154 in response to the comment from Reviewer 1, which show the inhibition of synergistic cytotoxicity by the perforin inhibitor, CMA. Thus, the accuracy of the results had could be increased. Thank you for your constructive suggestion.

Round 2

Reviewer 1 Report

Regarding author's respones: We appreciate your suggestions and completely agree with them because CDC and ADPC are the other important mechanisms of cytotoxicity elicited by daratumumab. Although we attempted to investigate these aspects, we could not complete the experiments due to shortage of time. ADCC and venetoclax elicit cytotoxicity via a mitochondria-dependent apoptosis pathway, whereas CDC or ADPC cytotoxicity is mitochondria independent. It is considered that mechanisms of combination efficacy of ADCC and venetoclax are different from those of CDC and venetoclax or ADPC and venetoclax. Therefore, we focused on enhancement of ADCC by venetoclax in this study. We would like to report the impact of venetoclax on Dara-induced CDC and ADPC in another paper.

I feel it is esential to address CDC/ADCP, given that it is difficult to perform in vivo experiments.  This review cannot understand why authors think ADCC is mitochondria-dependent, and CDC/ADCP is mitochondira-independent.

Apoptotic tumor cells (by venetoclax) can be efficienctly phagocytosed by macorphages, so it is important to know whether  venetoclax can enhance ADCP.  CDC, like pore-foroming cell death by NK cells, trigger cell lysis.  Again, the impact of venetoclax on CDC will be important. 

It is disappoinitng that authors have not addressed critical points in these 3 months. 

Author Response

Response to Reviewer #1 Comments

We sincerely appreciate the valuable comments.

Point 1: I feel it is essential to address CDC/ADCP, given that it is difficult to perform in vivo experiments. This review cannot understand why authors think ADCC is mitochondria-dependent, and CDC/ADCP is mitochondria-independent. Apoptotic tumor cells (by venetoclax) can be efficiently phagocytosed by macrophages, so it is important to know whether venetoclax can enhance ADCP. CDC, like pore-forming cell death by NK cells, trigger cell lysis. Again, the impact of venetoclax on CDC will be important. It is disappointing that authors have not addressed critical points in these 3 months.

Response 1: Additional experiments were conducted to confirm whether venetoclax could enhance complement-dependent cytotoxicity (CDC) by daratumumab using two CD38-positive myeloma cell lines (KMS12PE [1] and KMS27 [2]), which were used in this study and had revealed daratumumab mediated antibody-dependent cellular cytotoxicity (ADCC). However, daratumumab did not cause CDC in either of the two myeloma cell lines before examining the CDC-enhancing effect of venetoclax, as shown in Supplementary Fig S1. These results may be supported by a previous report [3] that tumor cells in only half of patients with myeloma were found to be sensitive to daratumumab-induced CDC. Therefore, it remains unclear whether CDC and antibody-dependent cellular phagocytosis (ADCP) could be involved in the synergistic effect of daratumumab and venetoclax on myeloma cells; these have been added to the discussion as the limitation of this study. However, venetoclax specifically inhibits BCL-2 in mitochondria, causing caspase activation via the mitochondrial apoptotic pathway and exerting cytotoxic activity [4-6]. In ADCC by NK cells, the intracellular entry of granzyme through perforin pore formation induces caspase activation via the mitochondrial apoptotic pathway [7]. These suggest that the synergistic effect of venetoclax and ADCC by NK cells via daratumumab on myeloma cells observed in this study may have been induced by activation of the mitochondrial apoptotic pathway. In contrast, the antitumor effects of CDC and ADCP are caused by direct cell-killing effect by lysis through complement pore formation and phagocytosis by macrophages, respectively, and the mitochondrial apoptosis pathway is not directly involved in either case [8-11]. These suggest that venetoclax is unlikely to enhance CDC or ADCP to myeloma cells via daratumumab. However, these are highly speculative and require further investigation in the future. These have been added to the discussion. In addition, this study showed that venetoclax only showed NK cell-mediated cytotoxic activity by daratumumab, and thus the title has been revised as follows: Synergistic effects of venetoclax and daratumumab on antibody-dependent cell-mediated natural killer cell cytotoxicity in multiple myeloma.

References

  1. Namba, M.; Ohtsuki, T.; Mori, M.; Togawa, A.; Wada, H.; Sugihara, T.; Yawata, Y.; Kimoto, T., Establishment of five human myeloma cell lines. In Vitro Cell Dev Biol 1989, 25, (8), 723-9.
  2. Otsuki, T.; Sakaguchi, H.; Hatayama, T.; Wu, P.; Takata, A.; Hyodoh, F., Effects of all-trans retinoic acid (ATRA) on human myeloma cells. Leuk Lymphoma 2003, 44, (10), 1651-6.
  3. de Weers, M.; Tai, Y. T.; van der Veer, M. S.; Bakker, J. M.; Vink, T.; Jacobs, D. C.; Oomen, L. A.; Peipp, M.; Valerius, T.; Slootstra, J. W.; Mutis, T.; Bleeker, W. K.; Anderson, K. C.; Lokhorst, H. M.; van de Winkel, J. G.; Parren, P. W., Daratumumab, a novel therapeutic human CD38 monoclonal antibody, induces killing of multiple myeloma and other hematological tumors. J Immunol 2011, 186, (3), 1840-8.
  4. Pan, R.; Hogdal, L. J.; Benito, J. M.; Bucci, D.; Han, L.; Borthakur, G.; Cortes, J.; DeAngelo, D. J.; Debose, L.; Mu, H.; Dohner, H.; Gaidzik, V. I.; Galinsky, I.; Golfman, L. S.; Haferlach, T.; Harutyunyan, K. G.; Hu, J.; Leverson, J. D.; Marcucci, G.; Muschen, M.; Newman, R.; Park, E.; Ruvolo, P. P.; Ruvolo, V.; Ryan, J.; Schindela, S.; Zweidler-McKay, P.; Stone, R. M.; Kantarjian, H.; Andreeff, M.; Konopleva, M.; Letai, A. G., Selective BCL-2 inhibition by ABT-199 causes on-target cell death in acute myeloid leukemia. Cancer Discov 2014, 4, (3), 362-75.
  5. Joza, N.; Susin, S. A.; Daugas, E.; Stanford, W. L.; Cho, S. K.; Li, C. Y.; Sasaki, T.; Elia, A. J.; Cheng, H. Y.; Ravagnan, L.; Ferri, K. F.; Zamzami, N.; Wakeham, A.; Hakem, R.; Yoshida, H.; Kong, Y. Y.; Mak, T. W.; Zuniga-Pflucker, J. C.; Kroemer, G.; Penninger, J. M., Essential role of the mitochondrial apoptosis-inducing factor in programmed cell death. Nature 2001, 410, (6828), 549-54.
  6. Di Martino, L.; Tosello, V.; Peroni, E.; Piovan, E., Insights on Metabolic Reprogramming and Its Therapeutic Potential in Acute Leukemia. Int J Mol Sci 2021, 22, (16), 8738.
  7. Martinez-Lostao, L.; Anel, A.; Pardo, J., How Do Cytotoxic Lymphocytes Kill Cancer Cells? Clin Cancer Res 2015, 21, (22), 5047-56.
  8. Golay, J.; Taylor, R. P., The Role of Complement in the Mechanism of Action of Therapeutic Anti-Cancer mAbs. Antibodies (Basel) 2020, 9, (4).
  9. Gul, N.; van Egmond, M., Antibody-Dependent Phagocytosis of Tumor Cells by Macrophages: A Potent Effector Mechanism of Monoclonal Antibody Therapy of Cancer. Cancer Res 2015, 75, (23), 5008-13.
  10. Young, J. D.; Ko, S. S.; Cohn, Z. A., The increase in intracellular free calcium associated with IgG gamma 2b/gamma 1 Fc receptor-ligand interactions: role in phagocytosis. Proc Natl Acad Sci U S A 1984, 81, (17), 5430-4.
  11. Markiewski, M. M.; Lambris, J. D., Is complement good or bad for cancer patients? A new perspective on an old dilemma. Trends Immunol 2009, 30, (6), 286-92.

Reviewer 2 Report

Overall, the manuscript is improved and the authors answered all of previous questions or comments. There is still few minor points:

In the results section, the authors describe expression pattern of BCL2 family proteins (In figure 2). However, since they are proteins, it should be stated that these are protein levels, not expression, which is rather used to describe gene expression.

Figure 3, it is not clear to me, why authors present these data in such a non-standard way- For all cell lines, there should be viability of cells at 0 nM concentration of venetoclax presented as 100% and then the treatment points always normalized to untreated cells. In this way, all of the cell lines will start at the same point in the graph, not in the middle.

Author Response

Response to Reviewer #2 Comments

We sincerely appreciate the encouraging comments.

Point 1: Overall, the manuscript is improved and the authors answered all of previous questions or comments. There is still few minor points: In the results section, the authors describe expression pattern of BCL2 family proteins (In figure 2). However, since they are proteins, it should be stated that these are protein levels, not expression, which is rather used to describe gene expression.

Response 1: According to kind observations by the reviewer, we have corrected “expression” in 2.2. Expression levels of BCL2 family proteins in MM cell lines to “protein level”

Point 2: Figure 3, it is not clear to me, why authors present these data in such a non-standard way- For all cell lines, there should be viability of cells at 0 nM concentration of venetoclax presented as 100% and then the treatment points always normalized to untreated cells. In this way, all of the cell lines will start at the same point in the graph, not in the middle.

Response 2: As the reviewer pointed out, the original Figure 3 seemed confusing. Figure 3 shows the susceptibility of cell lines to venetoclax in the concentration range of 0.39-100 nM for KMS12PE and KMS27 and 39-10000 nM for KM5 and U266, respectively. As a result, Figure 3 shows the viability rate at each point as 100% at 0 nM. Therefore, we have modified the results for the experiment using KMS12PE and KMS27, with the start point being 0.39 nM, and the results of the experiment using KM5 and U266, with the start point being 39 nM (in the middle of the graph). In addition, “The viability rate for each point is shown as 100% at 0 nM” has been added to the legend in Figure 3.

Reviewer 3 Report

The manuscript has been reasonably improved and most of my comments properly addressed.

Author Response

Response to Reviewer #3 Comments

We sincerely appreciate the encouraging comments.

Point 1: The manuscript has been reasonably improved and most of my comments properly addressed.

Response 1: We are grateful for your understanding and support.

Round 3

Reviewer 1 Report

Authors have addressed concerns from this reviewer.